# Exploiting Local and Global Structure for Point Cloud Semantic Segmentation with Contextual Point Representations

**Xu Wang**
College of Computer Science
and Software Engineering
Shenzhen University
Shenzhen, China
wangxu@szu.edu.cn

**Jingming He**
College of Computer Science
and Software Engineering
Shenzhen University
Shenzhen, China
hejingming519@gmail.com

**Lin Ma**[*]
Tencent AI Lab
Shenzhen, China
forest.linma@gmail.com

## Abstract

In this paper, we propose one novel model for point cloud semantic segmentation, which exploits both the local and global structures within the point cloud based on the contextual point representations. Specifically, we enrich each point representation by performing one novel gated fusion on the point itself and its contextual points. Afterwards, based on the enriched representation, we propose one novel graph pointnet module, relying on the graph attention block to dynamically compose and update each point representation within the local point cloud structure. Finally, we resort to the spatial-wise and channel-wise attention strategies to exploit the point cloud global structure and thereby yield the resulting semantic label for each point. Extensive results on the public point cloud databases, namely the S3DIS and ScanNet datasets, demonstrate the effectiveness of our proposed model, outperforming the state-of-the-art approaches. Our code for this paper is available at https://github.com/fly519/ELGS.

## 1 Introduction

The point cloud captured by 3D scanners has attracted more and more research interests, especially for the point cloud understanding tasks, including the 3D object classification [13, 14, 10, 11], 3D object detection [21, 27], and 3D semantic segmentation [25, 13, 14, 23, 10]. 3D semantic segmentation, aiming at providing class labels for each point in the 3D space, is a prevalent challenging problem. First, the points captured by the 3D scanners are usually sparse, which hinders the design of one effective and efficient deep model for semantic segmentation. Second, the points always appear unstructured and unordered. As such, the relationship between the points is hard to be captured and modeled.

As points are not in a regular format, some existing approaches first transform the point clouds into regular 3D voxel grids or collections of images, and then feed them into traditional convolutional neural network (CNN) to yield the resulting semantic segmentation [25, 5, 22]. Such a transformation process can somehow capture the structure information of the points and thereby exploit their relationships. However, such approaches, especially in the format of 3D volumetric data, require high memory and computation cost. Recently, another thread of deep learning architectures on point clouds, namely PointNet [13] and PointNet++ [14], is proposed to handle the points in an efficient and effective way. Specifically, PointNet learns a spatial encoding of each point and then aggregates all individual point features as one global representation. However, PointNet does not consider the

---

[*]Corresponding author.

local structures. In order to further exploit the local structures, PointNet++ processes a set of points in a hierarchical manner. Specifically, the points are partitioned into overlapping local regions to capture the fine geometric structures. And the obtained local features are further aggregated into larger units to generate higher level features until the global representation is obtained. Although promising results have been achieved on the public datasets, there still remains some opening issues.

First, each point is characterized by its own coordinate information and extra attribute values, *i.e.* color, normal, reflectance, etc. Such representation only expresses the physical meaning of the point itself, which does not consider its neighbouring and contextual ones. Second, we argue that the local structures within point cloud are complicated, while the simple partitioning process in PointNet++ cannot effectively capture such complicated relationships. Third, each point labeling not only depends on its own representation, but also relates to the other points. Although the global representation is obtained in PointNet and PointNet++, the complicated global relationships within the point cloud have not been explicitly exploited and characterized.

In this paper, we propose one novel model for point cloud semantic segmentation. First, for each point, we construct one contextual representation by considering its neighboring points to enrich its semantic meaning by one novel gated fusion strategy. Based on the enriched semantic representations, we propose one novel graph pointnet module (GPM), which relies on one graph attention block (GAB) to compose and update the feature representation of each point within the local structure. Multiple GPMs can be stacked together to generate the compact representation of the point cloud. Finally, the global point cloud structure is exploited by the spatial-wise and channel-wise attention strategies to generate the semantic label for each point.

## 2 Related Work

Recently, deep models have demonstrated the feature learning abilities on computer vision tasks with regular data structure. However, due to the limitation of data representation method, there are still many challenges for 3D point cloud task, which is of irregular data structures. According to the 3D data representation methods, existing approaches can be roughly categorized as 3D voxel-based [5, 25, 22, 7, 9], multiview-based [18, 12], and set-based approaches [13, 14].

**3D Voxel-based Approach.** The 3D voxel-based methods first transform the point clouds into regular 3D voxel grids, and then the 3D CNN can be directly applied similarly as the image or video. Wu *et al.* [25] propose the full-voxels based 3D ShapeNets network to store and process 3D data. Due to the constraints of representation resolution, information loss are inevitable during the discretization process. Meanwhile, the memory and computational consumption are increases dramatically with respect to the resolution of voxel. Recently, Oct-Net [16], Kd-Net [7], and O-CNN [22] have been proposed to reduce the computational cost by skipping the operations on empty voxels.

**Multiview-based Approach.** The multiview-based methods need to render multiple images from the target point cloud based on different view angle settings. Afterwards, each image can be processed by the traditional 2D CNN operations [18]. Recently, the multiview image CNN [12] has been applied to 3D shape segmentation, and has obtained satisfactory results. The multiview-based approaches help reducing the computational cost and running memory. However, converting the 3D point cloud into images also introduce information loss. And how to determine the number of views and how to allocate the view to better represent the 3D shape still remains as an intractable problem.

**Set-based Approach.** PointNet [13] is the first set-based method, which learns the representation directly on the unordered and unstructured point clouds. PointNet++ [14] relies on the hierarchical learning strategy to extend PointNet for capturing local structures information. PointCNN [10] is further proposed to exploit the canonical order of points for local context information extraction.

Recently, there have been several attempts in the literature to model the point cloud as structured graphs. For example, Qi *et al.* [15] propose to build a $k$-nearest neighbor directed graph on top of point cloud to boost the performance on the semantic segmentation task. SPGraph [8] is proposed to deal with large scale point clouds. The points are adaptively partitioned into geometrically homogeneous elements to build a superpoint graph, which is then fed into a graph convolutional network (GCN) for predicting the semantic labels. DGCNN [24] relies on the edge convolution operation to dynamically capture the local shapes. RS-CNN [11] extends regular grid CNN to irregular configuration, which encodes the geometric relation of points to achieve contextual shape-aware learning of point cloud.

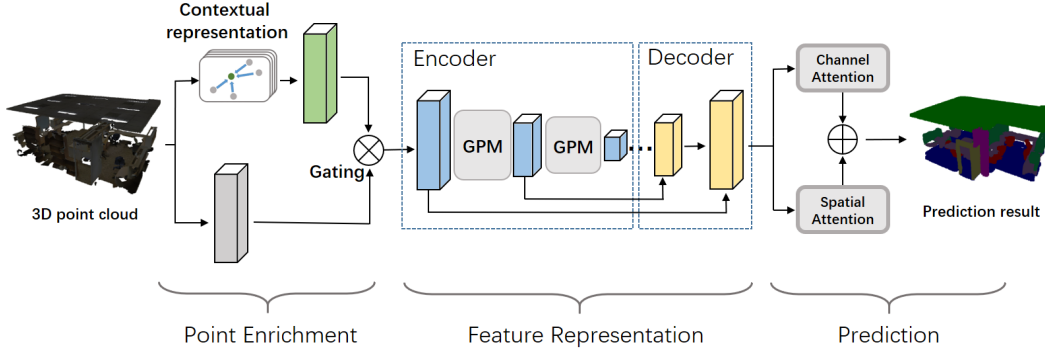

Figure 1: Our proposed model for the point cloud semantic segmentation, consisting of three fully-coupled components. The point enrichment not only considers the point itself but also its contextual points to enrich the corresponding semantic representation. The feature representation relies on conventional encoder-decoder architecture with lateral connections to learn the feature representation for each point. Specifically, the GPM is proposed to dynamically compose and update each point representation via a GAB module. For the prediction, we resort to both channel-wise and spatial-wise attentions to exploit the global structure for the final semantic label prediction for each point.

These approaches mainly focus on the point local relationship exploitation, and neglect the global relationship.

Unlike previous set-based methods that only consider the raw coordinate and attribute information of each single point, we pay more attentions on the spatial context information within neighbor points. Our proposed context representation is able to express more fine-grained structural information. We also rely on one novel graph pointnet module to compose and update each point representation within the local point cloud structure. Moreover, the point cloud global structure information is considered with the spatial-wise and channel-wise attention strategies.

## 3 Approach

The point cloud semantic segmentation aims to take the 3D point cloud as input and assign one semantic class label for each point. We propose one novel model for handling this point cloud semantic segmentation, as shown in Fig. 1. Specifically, our proposed network consists of three components, namely, the point enrichment, the feature representation, and the prediction. These three components fully couple together, ensuring an end-to-end training manner.

**Point Enrichment.** To make accurate class prediction for each point within the complicated point cloud structure, we need to not only consider the information of each point itself but also its neighboring or contextual points. Different from the existing approaches, relying on the information of each point itself, such as the geometry, color, etc., we propose one novel point enrichment layer to enrich each point representation by taking its neighboring or contextual points into consideration. With the incorporated contextual information, each point is able to sense the complicated point cloud structure information. As will be demonstrated in Sec. 4.4, the contextual information, enriching the semantic information of each point, can help boosting the final segmentation performance.

**Feature Representation.** With the enriched point representation, we resort to the conventional encoder-decoder architecture with lateral connections to learn the feature representation for each point. To further exploit local structure information of the point cloud, the GPM is employed in the encoder, which relies on the GAB to dynamically compose and update the feature representation of each point within its local regions. The decoder with lateral connections works on the compacted representation obtained from the encoder, to generate the semantic feature representation for each point.

**Prediction.** Based on the obtained semantic representations, we resort to both the channel-wise and spatial-wise attentions to further exploit the global structure of the point cloud. Afterwards, the semantic label is predicted for each point.

## 3.1 Point Enrichment

The raw representation of each point is usually its 3D position and associated attributes, such as color, reflectance, surface normal, etc. Existing approaches usually directly take such representation as input, neglecting its neighboring or contextual information, which is believed to play an essential role [17] for characterizing the point cloud structure, especially from the local perspective. In this paper, besides the point itself, we incorporate its neighboring points as its contextual information to enrich the point semantic representation. With such incorporated contextual information, each point is aware of the complicated point cloud structure information.

As illustrated in Fig. 2, a point cloud consists of $N$ points, which can be represented as $\{P_1, P_2, ..., P_N\}$, with $P_i \in \mathbb{R}^{C_f}$ denoting the attribute values of the $i$-th point, such as position coordinate, color, normal, etc. To characterize the contextual information for each point, $k$-nearest neighbor set $\mathcal{N}_i$ within the local region centered on $i$-th point are selected and concatenated together, where the contextual representation $R_i \in \mathbb{R}^{kC_f}$ of the given point $i$ is as follows:

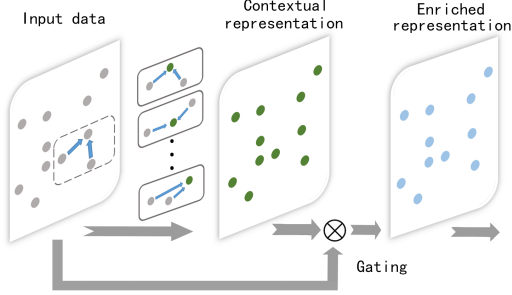

$$R_i = \underset{j \in \mathcal{N}_i}{\|} P_j. \tag{1}$$

Figure 2: The point enrichment process relies on our proposed gated fusion strategy to enrich the point representation by considering the neighbouring and contextual points of each point.

For each point, we have two different representations, specifically the $P_i$ and $R_i$. However, these two representations are of different dimensions and different characteristics. How to effectively fuse them together to produce one more representative feature for each point remains an open issue. In this paper, we propose a novel gated fusion strategy. We first feed $P_i$ into one fully-connected (FC) layer to obtain a new feature vector $\tilde{P}_i \in \mathbb{R}^{kC_f}$. Afterwards, the gated fusion operation is performed:

$$
\begin{aligned}
g_i &= \sigma(w_i R_i + b_i), & \hat{P}_i &= g_i \odot \tilde{P}_i, \\
g_i^R &= \sigma(w_i^R \tilde{P}_i + b_i^R), & \hat{R}_i &= g_i^R \odot R_i,
\end{aligned}
\tag{2}
$$

where $w_i, w_i^R \in \mathbb{R}^{kC_f \times kC_f}$ and $b_i, b_i^R \in \mathbb{R}^{kC_f}$ are the learnable parameters. $\sigma$ is the non-linear sigmoid function. $\odot$ is the element-wise multiplication. The gated fusion aims to mutually absorb useful and meaningful information of $P_i$ and $R_i$. And the interactions between $P_i$ and $R_i$ are updated, yielding $\hat{P}_i$ and $\hat{R}_i$. As such, the $i$-th point representation is then enriched by concatenating them together as $\hat{P}_i \parallel \hat{R}_i$. For easing the following introduction, we will re-use $P_i$ to denote the enriched representation of the $i$-th point.

## 3.2 Feature Representation

Based on the enriched point representation, we rely on one traditional encoder-decoder architecture with lateral connections to learn the feature representation of each point.

### 3.2.1 Encoder

Although the enriched point representation has somewhat considered the local structure information, the complicated relationships within points, especially from the local perspective need to be further exploited. In order to tackle this challenge, we propose one novel GPM in the encoder, which aims to learn the composition ability between points and thereby more effectively capture the local structural information within the point cloud.

**Graph Pointnet Module.** Same as [14], we first use the sampling and grouping layers to divide the point set into several local groups. Within each group, the GPM is used to exploit the local relationships between points, and thereby update the point representation by aggregating the point information within the local structure.

As illustrated in Fig. 3, the proposed GPM consists of one multi-layer perceptron (MLP) and GAB. The MLP in conventional PointNet [13] and PointNet++ [14] independently performs on each point to mine the information within the point itself, while neglects the correlations and relationships among the points. In order to more comprehensively exploit the point relationship, we rely on the GAB to aggregate the neighboring point representations and thereby updated the point representation.

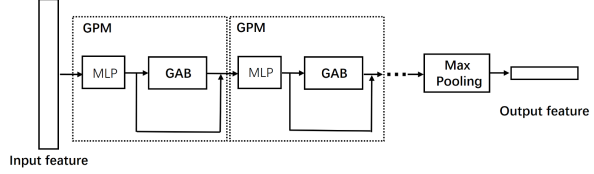

Figure 3: The architecture of our proposed GPM, which stacks MLP and GAB to exploit the point relationships within the local structure.

For each obtained local structure obtained by the sampling and grouping layers, GAB [20] first defines one fully connected undirected graph to measure the similarities between any two points with such local structures. Given the output feature map $\mathbf{G} \in \mathbb{R}^{C_e \times N_e}$ of the MLP layer in the GPM module, we first linearly project each point to one common space through a FC layer to obtain new feature map $\hat{\mathbf{G}} \in \mathbb{R}^{C_e \times N_e}$. The similarity $\alpha_{ij}$ between point $i$ and point $j$ is measured as follows:

$$\alpha_{ij} = \hat{G}_i \cdot \hat{G}_j. \tag{3}$$

Afterwards, we calculate the influence factor of point $j$ on point $i$:

$$\beta_{ij} = \text{softmax}_j(\text{LeakyReLU}(\alpha_{ij})), \tag{4}$$

where $\beta_{ij}$ is regarded as the normalized attentive weight, representing how point $j$ relates to point $i$. The representation of each point is updated by attentively aggregating the point representations with reference to $\beta_{ij}$:

$$\tilde{G}_i = \sum_{j=1}^{N_e} \beta_{ij} \hat{G}_j. \tag{5}$$

It can be observed that the GAB dynamically updates the local feature representation by referring to the similarities between points and captures their relationships. Moreover, in order to preserve the original information, the point feature after MLP is concatenated with the updated one via one skip connection through a gated fusion operation, as shown in Fig. 3.

Please note that we can stack multiple GPMs, as shown in Fig. 3, to further exploit the complicated non-linear relationships within each local structure. Afterwards, one max pooling layer is used to aggregate the feature map into a one-dimensional feature vector, which not only lowers the dimensionality of the representation, thus making it possible to quickly generate compact representation of the point cloud, but also help filtering out the unreliable noises.

### 3.2.2 Decoder

For decoder, we use the same architecture as [14]. Specifically, we progressively upsample the compact feature obtained from the encoder until the original resolution. Please note that for preserving the information generated in the encoder as much as possible, lateral connections are also used.

### 3.3 Prediction

After performing the feature representation, rich semantic representation for each point is obtained. Note that our previous operations, including contextual representation and feature representation, only mine the point local relationships. However, the global information is also important, which needs to be considered when determining the label for each individual point. For the semantic segmentation task, two points departing greatly in space may belong to the same semantic category, which can be jointly considered to mutually enhance their feature representations. Moreover, for high-dimensional feature representations, the inter-dependencies between feature channels also exist. As such, in order to capture the global context information for each point, we introduce two attention modules, namely spatial-wise and channel-wise attentions [4] for modeling the global relationships between points.

**Spatial-wise Attention**. To model rich global contextual relationships among points, the spatial-wise attention module is employed to adaptively aggregate spatial contexts of local features. Given the

feature map $\mathbf{F} \in \mathbb{R}^{C_d \times N_d}$ from the decoder, we first feed it into two FC layers to obtain two new feature maps $\mathbf{A}$ and $\mathbf{B}$, respectively, where $\{\mathbf{A}, \mathbf{B}\} \in \mathbb{R}^{C_d \times N_d}$. $N_d$ is the number of points and $C_d$ is number of feature channel. The normalized spatial-wise attentive weight $v_{ij}$ measures the influence factor of point $j$ on point $i$ as follows:

$$v_{ij} = \text{softmax}_j(A_i \cdot B_j), \tag{6}$$

Afterwards, the feature map $\mathbf{F}$ is fed into another FC layer to generate a new feature map $\mathbf{D} \in \mathbb{R}^{C_d \times N_d}$. The output feature map $\hat{\mathbf{F}} \in \mathbb{R}^{C_d \times N_d}$ after spatial-wise attention is obtained:

$$\hat{F}_i = \sum_{j=1}^{N_d}(v_{ij}D_j) + F_i. \tag{7}$$

As such, the global spatial structure information is attentively aggregated with each point representation.

**Channel-wise Attention**. The channel-wise attention performs similarly with the spatial-wise attention, with the channel attention map explicitly modeling the interdependencies between channels and thereby boosting the feature discriminability. Similar as the spatial-wise attention module, the output feature map $\tilde{\mathbf{F}} \in \mathbb{R}^{C_d \times N_d}$ is obtained by aggregating the global channel structure information with each channel representation.

After summing the feature maps $\hat{\mathbf{F}}$ and $\tilde{\mathbf{F}}$, the semantic label for each point can be obtained with one additional FC layer. With such attention processes from the global perspective, the feature representation of each point is updated. As such, the complicated relationships between the points can be comprehensively exploited, yielding more accurate segmentation results.

## 4 Experiment

### 4.1 Experiment Setting

**Dataset**. To evaluate the performance of proposed model and compare with state-of-the-art, we conduct experiments on two public available datasets, the Stanford 3D Indoor Semantics (S3DIS) Dataset [1] and ScanNet Dataset[2]. The S3DIS dataset comes from real scan of the indoor environment, including 3D scans of Matterport scanners from 6 areas. There are 271 rooms divided by room. ScanNet is a point cloud dataset with scanned indoor scenes. It has 22 categories of semantic tags, with 1513 scenes. ScanNet contains a wide variety of spaces. Each point is annotated with an instance-level semantic category label.

**Implementation Details**. The number of neighboring points $k$ in contextual representation is set as 3, where the farthest distance for neighboring point is fixed to 0.06. For feature extraction, a four-layer encoder is used, where the spatial scale of each layer is set as 1024, 256, 64, and 16, respectively. The GPM is enabled in the first two layers of the encoder, to exploit the local relationships between points. The maximum training epochs for S3DIS and ScanNet are set as 120 and 500, respectively.

**Evaluation Metric**. Two widely used metrics, namely overall accuracy (OA) and mean intersection of union (mIoU), are used to measure the semantic segmentation performance. OA is the prediction accuracy of all points. IoU measures the ratio of the area of overlap to the area of union between the ground truth and segmentation result. mIoU is the average of IoU over all categories.

**Competitor Methods**. For S3DIS dataset, we compare our method with PointNet [13], PointNet++ [14],

Table 1: Results of S3DIS dataset on "Area 5" and over 6 fold in terms of OA and mIoU. [†] and [‡] indicate that the PointNet performances are directly copied from [8] and [3], respectively. [*] indicates that the PointNet++ performances are produced with the publicly available code.

| Test Area | Method | OA | mIoU |
|---|---|---|---|
| Area5 | PointNet[†] [13] | - | 41.09 |
| | SEGCloud [19] | - | 48.92 |
| | RSNet [6] | - | 51.93 |
| | PointNet++[*] [14] | 86.43 | 54.98 |
| | SPGraph [8] | 86.38 | 58.04 |
| | Ours | **88.43** | **60.06** |
| 6 fold | PointNet[‡] [13] | 78.5 | 47.6 |
| | SGPN [23] | 80.8 | 50.4 |
| | Engelmann et al. [3] | 81.1 | 49.7 |
| | A-SCN [26] | 81.6 | 52.7 |
| | SPGraph [8] | 85.5 | 62.1 |
| | DGCNN [24] | 84.3 | 56.1 |
| | Ours | **87.6** | **66.3** |

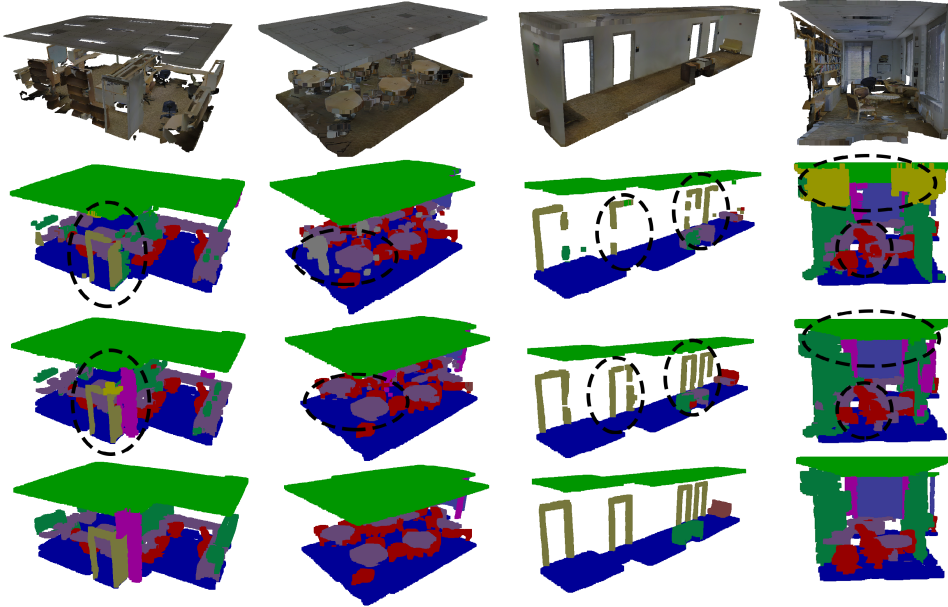

Figure 4: Qualitative results from the S3DIS dataset. All the walls are removed for better visualization. From top to bottom are the result of the Point Cloud, PointNet++, Ours, and Ground Truth, respectively. The segmentation results of our proposed model is closer to the ground truth than that of PointNet++.

SEGCloud [19], RSNet [6], SPGraph [8], SGPN [23], Engelmann *et al.* [3], A-SCN [26] and DGCNN [24]. For ScanNet dataset, we compare with 3DCNN [2], PointNet [13], PointNet++ [14], RSNet [6] and PointCNN [10].

Table 2: The segmentation results of S3DIS dataset in terms of IoU for each category.

| Test Area | Method | ceiling | floor | wall | beam | column | window | door | table | chair | sofa | bookcase | board | clutter |
|---|---|---|---|---|---|---|---|---|---|---|---|---|---|---|
| Area5 | PointNet [13]in [8] | 88.80 | 97.33 | 69.80 | **0.05** | 3.92 | 46.26 | 10.76 | 52.61 | 58.93 | 40.28 | 5.85 | 26.38 | 33.22 |
| | SEGCloud [19] | 90.06 | 96.05 | 69.86 | 0.00 | 18.37 | 38.35 | 23.12 | 75.89 | 70.40 | 58.42 | 40.88 | 12.96 | 41.60 |
| | RSNet [6] | **93.34** | 98.36 | 79.18 | 0.00 | 15.75 | 45.37 | 50.10 | 65.52 | 67.87 | 22.45 | 52.45 | 41.02 | 43.64 |
| | PointNet++ [14] | 91.41 | 97.92 | 69.45 | 0.00 | 16.27 | 66.13 | 14.48 | 72.32 | 81.10 | 35.12 | 59.67 | **59.45** | 51.42 |
| | SPGraph [8] | 89.35 | 96.87 | **78.12** | 0.00 | **42.81** | 48.93 | **61.58** | **84.66** | 75.41 | **69.84** | 52.60 | 2.10 | 52.22 |
| | Ours | 92.80 | **98.48** | 72.65 | 0.01 | 32.42 | **68.12** | 28.79 | 74.91 | **85.12** | 55.89 | **64.93** | 47.74 | **58.22** |
| 6fold | PointNet [13] in [3] | 88.0 | 88.7 | 69.3 | 42.4 | 23.1 | 47.5 | 51.6 | 42.0 | 54.1 | 38.2 | 9.6 | 29.4 | 35.2 |
| | Engelmann *et al.* [3] | 90.3 | 92.1 | 67.9 | 44.7 | 24.2 | 52.3 | 51.2 | 47.4 | 58.1 | 39.0 | 6.9 | 30.0 | 41.9 |
| | SPGraph [8] | 89.9 | 95.1 | 76.4 | **62.8** | **47.1** | 55.3 | 68.4 | **73.5** | 69.2 | **63.2** | 45.9 | 8.7 | 52.9 |
| | Ours | **93.7** | **95.6** | **76.9** | 42.6 | 46.7 | **63.9** | **69.0** | 70.1 | **76.0** | 52.8 | **57.2** | **54.8** | **62.5** |

## 4.2 S3DIS Semantic Segmentation

We perform semantic segmentation experiments on the S3DIS dataset to evaluate our performance in indoor real-world scene scans and perform ablation experiments on this dataset. Same as the experimental setup in PointNet [13], we divide each room evenly into several $1m^3$ cube, with each uniformly sampleing 4096 points.

Same as [13, 3, 8], we perform 6-fold cross validation with micro-averaging. In order to compare with more methods, we also report the performance on the fifth fold only (Area 5). The OA and mIoU results are summarized in Table 1. From the results we can see that our algorithm performs better than other competitor methods in terms of both OA and mIoU metrics.

Besides, the IoU values of each category are summarized in Table 2, it can be observed that our proposed method achieves the best performance for several categories. For simple shapes such as "floor" and "ceiling", each model performs well, with our approach performing better. This is mainly due to that the prediction layer of our propose method incorporates the global structure information between points, which enhances the point representation in the flat area. For categories with complex local structure, such as "chair" and "bookcase", our model shows the best performance,

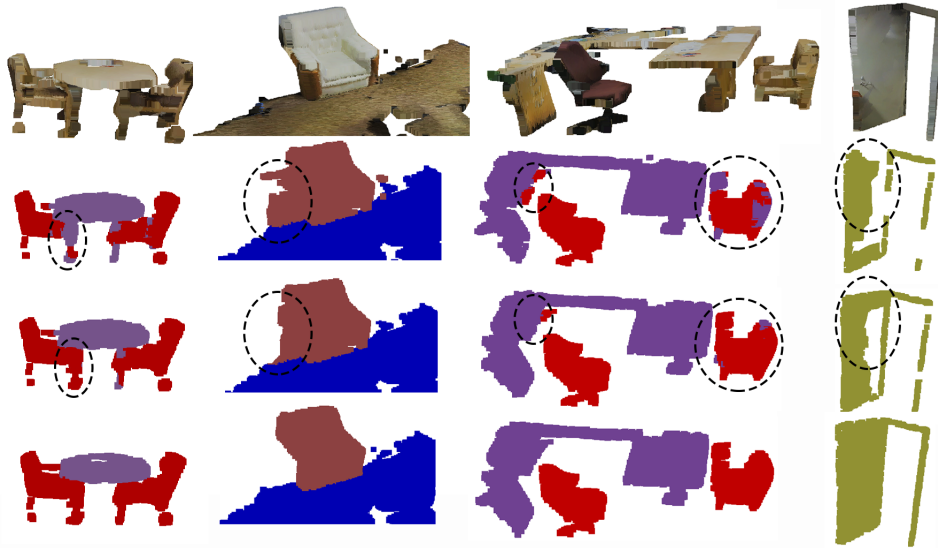

Figure 5: Qualitative results from the S3DIS dataset. From top to bottom are the result of the Point Cloud, PointNet++, Ours, and Ground Truth, respectively. The segmentation results of our proposed model is closer to the ground truth than that of PointNet++.

since we consider the contextual representation to enhance the relationship between each point and its neighbors, and use the GPM module to exploit the local structure information. However, the "window" and "board" categories are more difficult to distinguish from the "wall", as they are close to the "wall" in position and appear similarly. The key to distinguishing them is to find subtle shape differences and detect the edges. It can be observed that our model performs well on the "window" and "board" categories. In order to further demonstrate the effectiveness of our model, some qualitative examples from S3DIS dataset are provided in Fig. 4 and Fig. 5, demonstrating that our model can yield more accurate segmentation results.

### 4.3 ScanNet Semantic Segmentation

For the ScanNet dataset, the number of scenes trained and tested is 1201 and 312, same as [14, 10]. We only use its XYZ coordinate information. The results are illustrated in Table 3. Compared with other competitive methods, our proposed model achieves better performance in terms of both the OA and mIoU metrics.

Table 3: The segmentation results of ScanNet dataset in terms of both OA and mIoU.

| Method | OA | mIoU |
|---|---|---|
| 3DCNN [2] | 73.0 | - |
| PointNet [13] | 73.9 | - |
| PointNet++ [14] | 84.5 | 38.28 |
| RSNet [6] | - | 39.35 |
| PointCNN [10] | 85.1 | - |
| Ours | **85.3** | **40.6** |

### 4.4 Ablation Study

To validate the contribution of each module in our framework, we conduct ablation studies to demonstrate their effectiveness. Detailed experimental results are provided in Table 4.

**Contextual Representation Module**. After removing the contextual representation module in the input layer (denoted as w/o CR), we can see that the mIoU value dropped from 60.06 to 56.15, as shown in Table 4. Based on the results of each category in Table 5, some categories have significant drops in IoU, such as "column", "sofa", and "door". The contextual representation can enhance the point feature of the categories with complex local structures. We also replace the gating operation in the contextual representation with a simple concatenation operation. Due to the inequality of the two kinds of information,

Table 4: Ablation studies in terms of OA and mIoU.

| Method | OA | mean IoU |
|---|---|---|
| Ours(w/o CR) | 87.91 | 56.15 |
| Ours(w/o GPM) | 87.74 | 57.84 |
| Ours(w/o AM) | 87.90 | 58.67 |
| Ours(CR with concatenation) | 88.21 | 59.14 |
| Ours | **88.43** | **60.06** |

Table 5: Ablation studies and analysis in terms of IoU for each category.

| Method | ceiling | floor | wall | beam | column | window | door | table | chair | sofa | bookcase | board | clutter |
|--------|---------|-------|------|------|--------|--------|------|-------|-------|------|----------|-------|---------|
| Ours(w/o CR) | 92.62 | 98.69 | 69.65 | 0.00 | 7.81 | 66.02 | 21.92 | 74.64 | 84.38 | 29.94 | 62.53 | 66.52 | 55.19 |
| Ours(w/o AM) | 92.30 | 97.91 | 70.98 | 0.00 | 21.40 | 65.43 | 31.58 | 75.16 | 83.26 | 48.80 | 62.68 | 56.84 | 56.45 |
| Ours(w/o GPM) | 92.17 | 98.75 | 72.29 | 0.00 | 14.89 | 72.30 | 19.70 | 75.78 | 84.61 | 36.48 | 62.73 | 68.01 | 54.25 |
| Ours | 92.80 | 98.48 | 72.65 | 0.01 | 32.42 | 68.12 | 28.79 | 74.91 | 85.12 | 55.89 | 64.93 | 47.74 | 58.22 |

the OA and mIoU decreases. Thus, the proposed gating operation is useful for fusing the information of the point itself and its neighborhood.

**Graph Pointnet Module**. The segmentation performance of our model without GPM module (denoted as w/o GPM) also significantly drops, which indicates that both the proposed GPM and CR are important for performance improvement. Specifically, without GPM, the mIoU of the categories, such as "column" and "sofa" drops significantly.

**Attention Module**. Removing the attention module (denoted as w/o AM) decreases both OA and mIoU. Moreover, the performances on categories with large flat area, such as "ceiling", "floor", "wall", and "window", significantly drop. As aforementioned, the attention module aims to mine the global relationship between points. Two points within the same category may with large spatial distance. With the attention module, the features of these points are mutually aggregated.

We further incorporate the proposed CR, AM, and GPM together with DGCNN [24] for point cloud semantic segmentation, with the performances illustrated in Table 6. It can be observed that CR, AM, and GPM can help improving the performances, demonstrating the effectiveness of each module.

Table 6: Performances of DGCNN with our proposed module in terms of OA.

| Model | OA |
|-------|-----|
| DGCNN | 84.31 |
| DGCNN+CR | 85.35 |
| DGCNN+GPM | 84.90 |
| DGCNN+AM | 85.17 |
| DGCNN+CR+GPM+AM | 86.07 |

**Model Complexity**. Table 7 illustrates the model complexity comparisons. The sample sizes for all the models are fixed as 4096. It can be observed that the inference time of our model (28ms) is less than the other competitor models, except for PointNet (5.3ms) and PointNet++ (24ms). And the model size seems to be identical with other models except PointCNN, which presents the largest model.

**Robustness under Noise**. We further demonstrate the robustness of our proposed model with respect to PointNet++. As for scaling, when the scaling ratio are $50\%$, the OA of our proposed model and PointNet++ on segmentation task decreases by $3.0\%$ and $4.5\%$, respectively. As for rotation, when the rotation angle is $\frac{\pi}{10}$, the OA of our proposed model and PointNet++ on segmentation task decreases by $1.7\%$ and $1.0\%$, respectively. As such, our model is more robust to scaling while less robust to rotation.

Table 7: Model complexity

| Model | Time (ms) | Size (M) |
|-------|-----------|----------|
| PointNet | 5.3 | 1.17 |
| DGCNN | 42.0 | 0.99 |
| PointNet++ | 24.0 | 0.97 |
| RSNet | 60.4 | 6.92 |
| PointCNN | 34.4 | 11.51 |
| Ours | 28.0 | 1.04 |

## 5  Conclusion

In this paper, we proposed one novel network for point cloud semantic segmentation. Different with existing approaches, we enrich each point representation by incorporating its neighboring and contextual points. Moreover, we proposed one novel graph pointnet module to exploit the point cloud local structure, and rely on the spatial-wise and channel-wise attention strategies to exploit the point cloud global structure. Extensive experiments on two public point cloud semantic segmentation datasets demonstrating the superiority of our proposed model.

**Acknowledgments**

This work was supported in part by the National Natural Science Foundation of China (Grant 61871270 and Grant 61672443), in part by the Natural Science Foundation of SZU (grant no. 827000144) and in part by the National Engineering Laboratory for Big Data System Computing Technology of China.

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
