[Reviews · NeurIPS 2019]

Reviewer 1



1. Originality: The method is a combination of existing techniques, the attention has been well explored in GNN, which solves a simiilar to the point cloud analysis. Actually, point could is a kind of graph data. The contextual representation is just a fusion of neighboring features with the central one, which is quite straightforward. Meanwhile, the choice of fusion operations (equ 2,5,7) is not well explained and motivated. Local information has been explored in the point cloud community. Some related works are not cited and discussed. For example, Dynamic Graph CNN for Learning on Point Clouds, Relation-Shape Convolutional Neural Network for Point Cloud Analysis, they all offically accepted by peer-reviewed journals/conferences, and have papers on arxiv before NeuIPS submission deadline. Mining Point Cloud Local Structures by Kernel Correlation and Graph Pooling Pointwise Convolutional Neural Networks 2. Quality: The paper is technically sound. Some designs are not well supported by experiments or not well motivated, as I mentioned above. No results on running time complexity. 3. Clarity: The paper is well written and easy to follow. 4. Significance: The paper is incremental to previous work. While considering both global and local information is benefical for point cloud segmentation, to my opinion, the method does not show its advantages over previous work. For example, both the DGCNN and PointNet++ incorporates contextual information to enrich the point feature either by feature space(DGCNN) or geometric space(PointNet++, if we do not use the sampling to reduce the point number in the original PointNet++, it is a kind of feature extractor by considering neighboring information), the method does not show its advantage over these method. Actually, the method only shows that by combining many factors it can achieve better performance, however, it is unclear what will happen if we replace the proposed module with the existing technique. This could be more interest to readers to know the advance of the proposed method.

Reviewer 2



The authors present a novel network architecture and encoding for point clouds. Specifically, they propose to use additionally to the plain point position an enriched representation that includes the positions of the nearest neighbors of the point. The paper is technically sound, though I have some additional questions which I state below (see "Improvements"). The paper does not include a discussion of the failure modes of the proposed algorithm (especially the invariance properties w.r.t. rotation and scale could be interesting), but is reasonably well evaluated. The paper is clearly written and well-structured. I do find the described idea moderately interesting, where my main concerns are 1) general applicability of the method and 2) the failure modes of the proposed approach. I will go more in depth on these issues in the "Improvements" section. Additional comments after the rebuttal: I thank the authors for the additional insightful experiments and their detailed response. On ground of the classification scores, I tend to accept the paper. However, I still feel that there is not a good explanation for the globally optimized feature combination function.

Reviewer 3



The authors proposed to exploit the structure relationships between point clouds from both global and local perspectives are very enlightening. As such, the complicated relationships between point clouds can be more comprehensively exploited. Moreover, one novel contextual representation of each point is proposed, which considers its neighboring points to enrich the semantic meaning of each point. Such contextual representations are clearly motivated, with ablation studies demonstrating the corresponding contributions. The corresponding novelties and contributions have been summarized in the “Contributions” part. And the questions and some detailed comments are listed in the following. 1. I am wondering the results if considering the spatial-wise and channel-wise attention with each GPM. How does it perform, comparing with the proposed GPM. 2. For the ablation studies in Table 5, it seems that the performances of different components, namely CR, AM, and GPM, perform differently over different categories. Please provide more explanations. 3. What about the performances by stacking different numbers of GPMs? 4. Some more qualitative results should be provided.

[Author Response · NeurIPS 2019]

We sincerely thank all three reviewers for their valuable comments, with the following being our responses.

**R1&R2) Regarding the contextual representation.** We proposed one novel gated fusion strategy to mutually absorb useful and meaningful information of each point and its neighboring points to enrich its semantic representation. As in Eq. (2), $g_i^c$ and $g_i$ are determined by the representations of the each point and its neighboring ones. As such, although the weights in Eq. (2) are learned to be 'static', the enriched representation is adaptively determined by the point itself and its neighboring ones. Moreover, compared with the simple concatenation, the proposed gated fusion can effectively enrich the point representation yielding better performances, as illustrated in Table 4 (the submitted paper).

**R1&R2) Regarding the model complexity.** Table 1 (response letter) illustrates the model complexity comparisons. The sample sizes for all the models are fixed as 4096. It can be observed that the inference time of our model (28ms) is less than the other models, except for PointNet (5.3ms) and PointNet++ (24ms). And the model size seems to be identical with other models except PointCNN, which yields the largest model.

**R1) Regarding the advance of the proposed model.** Table 4 (the submitted paper) ablates the contribution of each component, namely CR, AM and GPM. We further incorporate the proposed CR, AM, and GPM together with DGCNN for point cloud semantic segmentation, with the performances illustrated in Table 2 (response letter). It can be observed that CR, AM, and GPM can help improving the performances, demonstrating the corresponding superiority. We will include such experiments in our revised paper.

**R1) Regarding missing related work.** Thanks for your suggestion. We will include the papers accordingly in our revised paper.

**R2) Regarding the effects of the order-specific weights:** Please note that the $k$ neighboring points as in Eq. (1) are randomly sampled within the point neighborhood. We are sorry for not providing clearer information in our manuscript. In order to examine the effects of the order-specific weights, we random shuffle the weights of the $k$ neighboring points in our CR module during the inference. The results with multiple inferences appear to be almost the same ($88.43\%\pm0.02\%$). Thus, the repositioning or reordering does not affect the corresponding performances.

**R2) Regarding the general limitations (KNN):** We agree that KNN encoding step as one general limitation makes it impossible for gradient propagation. We are considering to use self-attention to aggregate the features within the point neighborhood, which can thereby make the model suitable for other tasks besides the discriminative ones.

**R2) Regrading the performance on the classification task.** We evaluate our model on the ModelNet40 shape classification benchmark, shown in Table 3 (response letter). As usual, we uniformly sample 1024 points on mesh faces according to the face area and normalize them into a unit sphere. Only the coordinates of the sampled points are used, with the original meshes discarded. The results in Table 3 (response letter) clearly demonstrate the generalization ability of our proposed model, which achieves comparable performances with state-of-the-art models on classification task.

**R2) Regarding the robustness under noise in the data.** We demonstrate the robustness of our proposed model with respect to PointNet++. As for scaling, when the scaling ratio are $50\%$, the OA of our proposed model and PointNet++ on segmentation task decreases by $3.0\%$ and $4.5\%$, respectively. As for rotation, when the rotation angle is $\frac{\pi}{10}$, the OA of our proposed model and PointNet++ on segmentation task decreases by $1.7\%$ and $1.0\%$, respectively. As such, our model is more robust to scaling while less robust to rotation. We will include such discussions in our revised version.

**R3) Regarding spatial/channel-wise attentions with each GPM.** The performance of different GPM settings are summarized in Table 4 (response letter). The default GPM setting with spatial-wise attention achieves the best performance, where the channel-wise attention appears to decrease the performance.

**R3) Regarding the number of GPMs.** As shown in Table 4 (response letter), when stacking 3 GPMs, our proposed model achieves the best performance. Introducing more GPMs will increase the model capacity, resulting in performance improvement from 2 GPMs to 3 GPMs. Afterwards, with more GPMs stacked, more parameters are introduced, which cannot ensure an adequate training with limited data, resulting the performance degradation.

**R3) Regarding performances of different categories.** The CR module performs well on the categories with context dependency, *e.g.*, the category "column" always appears with "wall". Without CR module, the OA decreases by $24\%$. Both CR and GPM module are sensitive to local complicated structure information, *e.g.*, the OA on category "sofa" increases by $26\%$ and $19\%$, respectively. The AM aggregates global information and improve the performance of category with large area, *e.g.*, the OA on category "window" increases by $3\%$.

**R3) Regarding more qualitative results:** Thank you very much for the comment. We will include more qualitative results in our revised paper.

Table 1: Model complexity

| Model | Time (ms) | Size (M) |
|---|---|---|
| PointNet | 5.3 | 1.17 |
| DGCNN | 42.0 | 0.99 |
| PointNet++ | 24.0 | 0.97 |
| RSNet | 60.4 | 6.92 |
| PointCNN | 34.4 | 11.51 |
| Ours | 28.0 | 1.04 |

Table 2: DGCNN performance

| Model | OA |
|---|---|
| DGCNN | 84.31 |
| DGCNN+CR | 85.35 |
| DGCNN+GPM | 84.90 |
| DGCNN+AM | 85.17 |
| DGCNN+ALL | 86.07 |

Table 3: Classification results

| Model | acc. |
|---|---|
| Pointwise-CNN (CVPR'18) | 86.1 |
| PointNet (CVPR'16) | 89.2 |
| SCN (CVPR'18) | 90.0 |
| PointNet++ (NIPS'17) | 90.7 |
| KCNet (CVPR'18) | 91.0 |
| MRTNet (ECCV'18) | 91.2 |
| PointCNN (NeurIPS'18) | 91.7 |
| DGCNN (TOG'19) | 92.2 |
| Ours | 91.5 |

Table 4: GPM performances

| Model | OA |
|---|---|
| Channel | 88.08 |
| Channel+Spatial | 88.23 |
| Spatial | **88.43** |
| 2 GPMs | 87.58 |
| 3 GPMs | **88.43** |
| 4 GPMs | 87.48 |
| 5 GPMs | 87.54 |

[Meta-Review · NeurIPS 2019]

The paper introduced a new encoding of point clouds based on graph neural networks. It received scores of 8,6,8. The reviews for this paper were overly positive, and do not sufficiently convey the strengths of the paper to match the rating. The AC read the paper, rebuttal, and all reviews. The novelty of this paper is limited, particularly in light of missed relevant related work: Qi et al, 3D Graph Neural Networks for RGBD Semantic Segmentation, ICCV'19 This paper needs to be cited and discussed. Second, the authors need to improve the writing, in particular, make the writing slightly more scientific by omitting words such as "nowadays", and overly using the word "one". The AC will not overturn the reviewers' recommendation, however, urges the authors to polish their camera ready, and add discussion to relevant work.